# Dihydrotestosterone Augments the Angiogenic and Migratory Potential of Human Endothelial Progenitor Cells by an Androgen Receptor-Dependent Mechanism

**DOI:** 10.3390/ijms25094862

**Published:** 2024-04-29

**Authors:** Mirel Adrian Popa, Cristina Maria Mihai, Viorel Iulian Șuică, Felicia Antohe, Raghvendra K. Dubey, Brigitte Leeners, Maya Simionescu

**Affiliations:** 1Institute of Cellular Biology and Pathology “Nicolae Simionescu” of the Romanian Academy, 050568 Bucharest, Romania; mirel.popa@icbp.ro (M.A.P.); dr.corotchi@gmail.com (C.M.M.); viorel.suica@icbp.ro (V.I.Ș.); felicia.antohe@icbp.ro (F.A.); 2Department for Reproductive Endocrinology, University Zurich, 8006 Zürich, Switzerland; raghvendra.dubey@usz.ch (R.K.D.); brigitte.leeners@usz.ch (B.L.)

**Keywords:** endothelial progenitor cells, dihydrotestosterone, angiogenesis, migration

## Abstract

Endothelial progenitor cells (EPCs) play a critical role in cardiovascular regeneration. Enhancement of their native properties would be highly beneficial to ensuring the proper functioning of the cardiovascular system. As androgens have a positive effect on the cardiovascular system, we hypothesized that dihydrotestosterone (DHT) could also influence EPC-mediated repair processes. To evaluate this hypothesis, we investigated the effects of DHT on cultured human EPCs’ proliferation, viability, morphology, migration, angiogenesis, gene and protein expression, and ability to integrate into cardiac tissue. The results showed that DHT at different concentrations had no cytotoxic effect on EPCs, significantly enhanced the cell proliferation and viability and induces fast, androgen-receptor-dependent formation of capillary-like structures. DHT treatment of EPCs regulated gene expression of androgen receptors and the genes and proteins involved in cell migration and angiogenesis. Importantly, DHT stimulation promoted EPC migration and the cells’ ability to adhere and integrate into murine cardiac slices, suggesting it has a role in promoting tissue regeneration. Mass spectrometry analysis further highlighted the impact of DHT on EPCs’ functioning. In conclusion, DHT increases the proliferation, migration, and androgen-receptor-dependent angiogenesis of EPCs; enhances the cells’ secretion of key factors involved in angiogenesis; and significantly potentiates cellular integration into heart tissue. The data offer support for potential therapeutic applications of DHT in cardiovascular regeneration and repair processes.

## 1. Introduction

Endothelial progenitor cells (EPCs) are bone-marrow-derived cells originally reported in the peripheral circulation by Asahara et al. [1]. Studies suggest that EPCs could contribute to the regeneration of damaged vascular endothelium and to the neovascularization of ischemic lesions [2]. There are reports that in patients with acute coronary artery disease (CAD), circulating EPCs are low in number and their migratory function is impaired. The authors conclude that the number of circulating EPCs is inversely correlated with the number of risk factors for CAD [3]. Similarly, cardiovascular risk factors (i.e., hypertension, diabetes, hyperlipidemia, smoking, obesity, etc.) have a negative effect on the number and function of EPCs. Studies showed that in healthy men with different degrees of risk factors but no history of CAD, baseline EPC levels were inversely correlated with the number of risk factors (combined Framingham risk score). In addition, endothelial functionality, as assessed by flow-mediated dilation, is significantly correlated with the number of EPCs; this measure is a more useful indicator of vascular reactivity than any conventional risk factors [4].

In cardiovascular diseases (CVD), sex differences are relevant to diagnosis, pathophysiology, and treatment efficiency [5,6] There are data showing the role of androgen hormones on the functionality of EPCs [7,8,9]. However, there are several conflicting reports that do not support this notion [10,11]. EPCs are “sensitive” to levels of testosterone or its active metabolite, dihydrotestosterone (DHT); it is speculated that the mechanism involves the signaling pathway related to Egr1 or RhoA/ROCK1 [12,13]. The main functions of EPCs are to form new blood vessels or to regenerate damaged endothelium. These functions decline with age, making recovery after trauma/injury precarious or even impossible. Interestingly, aging is a condition that accentuates the decline in functionality of EPCs, and these phenomena also correlate with the decrease in the concentration of hormones in the body, making regenerative therapy challenging [14]. The use and successful use of EPCs for cardiac repair suffer from suboptimal utilization, and this is due to the role of unexplored mechanisms that would facilitate or improve their repair potential and clinical relevance. As angiogenesis, or the formation of new blood vessels, is a key to preventing scar-tissue formation and promoting tissue repair, we examined the direct effect of DHT and the role of androgen receptors on the interaction of EPCs with cardiac slices.

We designed experiments to explore the mechanism(s) by which DHT affects the migration, angiogenesis, and integration of human umbilical-blood-derived EPCs in heart tissue. Here, we report that DHT significantly increases the functionality of EPCs by an androgen-receptor-mediated mechanism. The results offer support for the potential of DHT therapy in cardiovascular repair processes.

## 2. Results

### 2.1. Characterization of EPCs Isolated from Human Umbilical Cord Blood

Mononuclear cells (MNCs) isolated from umbilical cord blood were exposed to endothelial growth factors added to the EGM2 media. After 10–14 days in culture, the colony-forming units acquired an epithelial-like phenotype. The isolated EPCs were subjected to thorough characterization, as previously reported [15]. Flow cytometric analysis revealed that these cells were significantly positive for endothelial-specific markers such as CD31, CD44, CD73, CD105, and CD144, while they were significantly negative for the hematopoietic cell markers CD45, CD90, and CD309. Moreover, the isolated cells exhibited functional characteristics typical of endothelial cells: they formed robust capillary-like tubes when cultured on Matrigel^®^ (Figure 1a,b) and avidly took up acetylated-LDL (Figure 1c,d). These well-characterized EPCs were used for further experiments.

### 2.2. DHT Increases EPCs Proliferation and Does Not Affect Cell Viability and Morphology

First, we determined whether DHT at 1, 30, 100 and 1000 nM concentration is cytotoxic to cells. The experiments showed that at these concentrations, DHT is not cytotoxic to cells (Figure 2A).

Secondly, we tested for cell proliferation and viability by MTT assay, real-time cell proliferation monitored with an xCELLigence system, and DNA quantification. The results demonstrated that compared to controls, at all concentrations used, DHT increased (~30%) the number of EPCs exposed to the hormone for 4 days (Figure 2B,C). To be in line with physiological conditions, we chose to use 30 nM DHT throughout further experiments. The proliferation of EPCs was found to be dependent upon the availability of androgen receptor (AR). This observation was particularly evident in the experiments in which 10 µM of flutamide, a selective antagonist of AR, was used. Flutamide effectively inhibits DHT binding to AR. These experiments revealed that the proliferation rate was directly correlated with the presence of the AR signaling pathway (Figure 2D).

### 2.3. DHT Stimulation of EPCs Induces Robust Formation of Capillary-like Structures through an Androgen-Receptor-Dependent Signaling Pathway

The formation of capillary-like network structures on Matrigel-coated wells is part of the functional characterization test for EPCs. Our experiments showed that stimulation of cultured EPCs with 30 nM DHT (for 8 h) increased the number of closed structures (nodes) compared to controls.

Exposure (45 min) of EPCs to flutamide the AR inhibitor, followed by stimulation of DHT decreased the capacity of EPCs to form nodes (Figure 2E-center, and Figure 2F). Digital analysis of the images obtained from the stimulation experiment revealed an increase of up to 50% in the rate of nodes formation in the presence of DHT. Moreover, the role of AR in this process was underscored by the inhibitory effect of flutamide, which resulted in the node formation rates comparable to those of the controls.

### 2.4. Exposure of EPCs to DHT Stimulates Regulation of Gene Expression for AR, for the Genes Involved in Cell Migration, and for Pro-Angiogenic Genes, VEGFR-2 and PlGF

The effect of DHT stimulation on regulation of EPCs gene expression was assessed by qRT-PCR. The experiments showed that after 4 days DHT induced a significant upregulation of mRNA levels for AR (≈4-fold), and for molecules such as matrix metalloproteases that are involved in cell migration, including EMMPRIN (≈1.5-fold), MMP-2 (≈2.2-fold) and MMP-9 (≈2.7-fold). Similarly, the genes that contribute to the angiogenic ability of EPCs, such as vascular endothelial growth factor receptor- 2 (VEGFR-2) and placental growth factor (PlGF) were upregulated with e ≈1.2 and ≈3.4-fold change, respectively (as depicted in Figure 3).

### 2.5. DHT Induces in EPCs a Significant Increase in the Protein Level of AR, VEGFR-2, PlGF, EMMPRIN and MMP-9

After analysis of the effect of DHT on specific gene expression, we analyzed the levels of the proteins AR, EMMPRIN, MMP-2, MMP-9, VEGFR-2, and PlGF by western blot assay. We detected that compared to unstimulated cells, the DHT-treated EPCs exhibited a significant increase in expression of AR (≈1.3-fold), EMMPRIN (≈1.5-fold), MMP-9 (≈1.8-fold), VEGFR-2 (≈1.1-fold), and PlGF (≈1.6-fold). In contrast, the MMP-2 level was not altered in DHT-stimulated EPCs compared to controls, a result that did not correlate with the increased gene expression. One explanation would be that differential regulation of transcription and translation, as well as other mechanisms, regulate MMP-2 protein expression. In contrast, DHT-induced changes in expression of the proteins AR, VEGFR-2, PlGF, EMMPRIN and MMP-9 paralleled and was in good agreement with the comparable increases in gene expression in stimulated EPCs (Figure 4).

### 2.6. DHT Increases Secretion of EMMPRIN, MMP-9, Angiogenin and VEGF upon Indirect Contact between Stimulated EPCs and Murine Ventricular Slices

The experimental results demonstrated that the concentrations of proteins released by DHT-stimulated EPCs were dependent on both the presence of ventricular sections and the duration of co-culture. For proteins such as EMMPRIN, VEGF, and angiogenin, the presence of heart sections was found to be critical in maintaining high protein levels. At 48 h, the concentration of secreted EMMPRIN increased up to ~66.7%, that of angiogenin increased up to ~88%, and that of VEGF increased up to ~82%. Interestingly, we found that the concentration of secreted MMP-9 was dependent on the presence of DHT and was not limited by the duration of exposure to ventricular sections (Figure 5).

### 2.7. Migration of DHT-Stimulated EPCs Is Dependent on the Presence of Androgen Receptors

To analyze the migration of EPCs, we employed an improved “scratch assay”. The migration rate was assessed in real time, employing an xCELLigence E-Plate containing the patented PP rectangle device (described in the Methods section). The results showed that the migration of DHT-stimulated EPCs toward the empty space of the well was significantly faster (~50%) compared to that of non-stimulated cells. Moreover, the migration of stimulated EPCs was directly correlated with the presence and activity of the AR, a feature revealed by the experiments in which flutamide was used. Blocking of the AR reduced the migration (the cell index) to control values (Figure 6).

### 2.8. Migration of EPCs towards Cardiac Tissue Is Prompted by the Presence of DHT and Dependent on the Existence of Androgen Receptors

To assess the effect of DHT on the migration of EPCs, xCELLigence’s CIM-Plate system was used. As shown in Figure 7, we found that the rate of migration of DHT-treated EPCs towards cardiac tissues was consistently higher than the rate of migration of untreated or vehicle-exposed cells (controls) (Figure 7A). According to the measurements of the cell index, after 20 h, the migration of stimulated EPCs was ~50% faster than that of controls.

To determine whether AR have a role in the migration of stimulated cells, we performed similar experiments in the presence of the AR antagonist flutamide. We detected that blockage of the AR reduced by ~50% the cell index values obtained for DHT-stimulated cells (Figure 7B).

### 2.9. DHT Stimulation of EPCs Increases Their Adherence and Integration into Cardiac Tissue

A direct co-culture model and two methodologies were employed to elucidate the impact of DHT on EPCs’ attachment and integration into murine heart slices.

For immunocytochemistry, sections of the heart slices previously incubated with DHT-stimulated EPCs were incubated with the anti-human-nucleus primary antibody and then with HRP-conjugated secondary antibody. After incubation, the peroxidase reaction and H&E staining clearly revealed the presence of human EPC nuclei (brown) and mouse cell nuclei (blue) (Figure 8A).

Morphometric analysis revealed a significant increase in the number of adherent and integrated DHT-treated EPCs compared to controls. The IgG isotype was employed as a control. The data validated the use of digital counting through ImageJ software (version 1.54f), demonstrated a ~30% increase in the integration of DHT-stimulated EPCs versus untreated cells. (Figure 8B).

To verify these findings, qRT-PCR was employed to quantify the number of human cells attached to/integrated into ventricular slices. Assessment of human DNA quantification validated that the total number of human EPCs adhering to and integrated into the cardiac sections was ~50% higher than that of unstimulated cells. (Figure 8C).

### 2.10. Regulation of Proteins ‘Abundance Involved in Cellular Migration and Integration in DHT-Treated EPCs as Demonstrated by Mass Spectrometry

Relative quantification experiments were based on precursor ion alignment and intensity comparisons between DHT-treated and control EPCs, as previously described [16]. A coefficient of variation of <30% between technical replicates was allowed. For further bioinformatic analysis, an additional filter was selected to allow only proteins quantified with at least two peptides/proteins. A false discovery rate (FDR) < 0.05 was used as the cutoff for the peptide target confidence. The protein inference and relative quantification bioinformatic analysis revealed that, out of a total of 2521 proteins identified, 160 proteins were statistically significantly downregulated, and 218 proteins were significantly downregulated upregulated by a factor of >1.5 (DHT-treated samples/control samples) (Figure 9A). To characterize the cellular components, biological processes, and molecular functions, a FunRich analysis was performed on the 2251 proteins identified in both vehicle-exposed (control) EPCs and DHT-treated EPCs. The analysis revealed that among the proteins with statistically significantly modified expression (378), ~34 proteins were associated with response to wound healing, ~22 were involved in extracellular matrix organization, and 14 were involved in angiogenesis (Figure 9B). In terms of biological processes, 9% of the proteins were involved in neutrophil degranulation, 6% in mRNA splicing via spliceosome, and 4% in nuclear-transcribed mRNA catabolic processes. In terms of molecular functions, 28% of the proteins were involved in RNA binding, 8% in cadherin binding, and 7% in GTP binding. These results provided valuable insights into the cellular components, biological processes, and molecular functions of the proteins identified in these experiments (Figure 10). By integrating information from public databases and literature sources, the first 20 upregulated proteins were grouped based on their functions and roles. These proteins were associated with cell signaling, communication, extracellular matrix adhesion, protein transport and localization, nucleic acid metabolism, metabolism, and immune function. Similarly, the first 20 down-regulated proteins were grouped according to their functions and roles. These proteins were associated with extracellular matrix and cell adhesion, protein trafficking and localization, metabolism, and protein proteolysis.

## 3. Discussion

Endothelial dysfunction is one of the main contributors to cardiovascular disease. Perturbations in metabolic and hemodynamic mechanisms expose the heart and vasculature to risk factors that compromise physiological cell repair and accelerate endothelial dysfunction [17]. It is safe to postulate that early replacement of dysfunctional or damaged endothelium may protect against CVD.

Regeneration of dysfunctional/damaged endothelium involves angiogenesis, a process in which local ECs from pre-existing blood vessels or distant/circulating endothelial progenitor cells participate. Strategies that can facilitate EC regeneration are of great clinical relevance in counteracting/treating CVD, in which damage to ECs is a highly characteristic feature. In the heart and in blood vessels, EPCs hold great promise because of their potential to replace damaged ECs. Although this approach is promising, the poor engraftment of EPCs in vivo has limited its therapeutic potential and has intensified efforts to identify molecules and mechanisms that may enhance the repair/regenerative potential of EPCs. To this end, we designed experiments to elucidate the interaction between EPCs and the myocardium. The search for factors, hormones or molecules that could enhance the native regenerative properties of EPCs is ongoing [18,19,20].

The main goal of the present study was to assess the effect of androgen hormones and their ligands on the functionality of human umbilical-cord-derived EPCs. The cells obtained from this source have important advantages: they have a higher capacity for proliferation and regeneration and are easily available in large quantities compared with peripheral blood EPCs. Moreover, these cells are less likely to be rejected by the immune system compared to circulating EPCs [21,22,23]. This study is the first to investigate the effect of DHT on EPCs. The novelty consists in the demonstration that DHT (1) induces proliferation and increases the capacity of EPCs to form capillary-like networks by an AR-dependent mechanism; (2) amplifies the migration capability of EPCs by increasing the gene and protein expression of EMMPRIN and MMP-9; (3) stimulates EPCs’ secretion of angiogenin, VEGF, and PlGF—key factors involved in angiogenesis-based regenerative processes; and, as shown by mass spectrometry; and (4) activates numerous proteins involved in the signaling response to injury, wound healing, and cellular integration.

Our results demonstrated that DHT significantly enhanced EPCs’ proliferation and migration and promoted the formation of capillary-like structures, which are crucial for neovascularization. These effects were mediated by the AR, as evidenced by the results showing the inhibitory effect on these processes of the AR antagonist flutamide. These data agree with the results of previous studies and extend them [7,24]. The AR-mediated enhancement of EPCs’ migration appears to involve the activation of signaling pathways that promote integrin-mediated adhesion and cytoskeletal rearrangement [25]

Interestingly, in vitro experiments highlighted the ability of EPCs to migrate toward and into heart ventricular slices. Migration of EPCs was significantly enhanced in cells cultured in the presence of DHT compared with unstimulated cells. In contrast, EPCs’ migration capacity was reduced in response to the inhibition of androgen receptors (up to 40%).

Importantly, this effect was further enhanced by the presence of heart slices, suggesting that factors generated by ischemic tissue could stimulate EPCs to increase production of MMPs. One can safely assume that the increase in proteolytic activity is a precursor for the removal of damaged scar tissue. The data are consistent with the results of our previous experiments on human Wharton’s Jelly-derived mesenchymal stromal cells (MSCs) [26], except that, in the present experiments, the DHT-induced increase in MMP-2 mRNA was not accompanied by enhanced protein expression in EPCs. It has been reported that the formation and activity of MMPs are controlled at several steps, such as genetic transcription and the conversion of latent enzymes into active forms; therefore, the lack of correlation between MMP-2 mRNA expression and protein levels is not an uncommon phenomenon [27].

Our findings indicate that DHT induces a significant increase in the expression of MMP-9 for up to 48 h, suggesting that it has a role in promoting the migratory state of EPCs. Additionally, DHT treatment increases the synthesis of EMMPRIN and VEGF proteins in EPCs, potentially regulating tissue remodeling through coordinated regulation of the cell matrix and cell migration. VEGF, a key mediator of the pro-angiogenic effects of MMPs, is upregulated by DHT, and the proteolytic action of MMPs facilitates the release of extracellular0matrix-bound VEGF, initiating angiogenesis and cell migration and growth. The DHT-mediated increase in the synthesis of the molecules described above suggests that androgens may activate mechanisms critical for initiating tissue migration and remodeling [26,28,29]. The spatio-temporal regulation of EMMPRIN, MMP-9, MMP-2, and VEGF underscores the essential role of proteolytic activity in cardiovascular remodeling [30]. Our results highlight the capacity of DHT to modulate EPCs’ functions and promote neovascularization.

In EPCs, DHT treatment enhanced cellular sprouting on Matrigel-coated wells, which is the precursor mechanism for angiogenesis and a hallmark of endothelial repair [22,31,32]. Interestingly, the sprouting effects of DHT were blocked by treatment with flutamide. AR expression in EPCs is upregulated by DHT. It is tempting to speculate that in patients with low hormone levels receiving EPC-based therapy, exposure of the cells to DHT prior to infusion could improve endothelial regeneration.

Regarding the mechanism(s) involved in angiogenesis, MMPs and VEGF facilitate cell migration and the formation of new blood vessels [33]. These findings indicate a spatio-temporal regulation of EMMPRIN, MMP-9, MMP-2, and VEGF that is consistent with ST-elevation myocardial infarction (STEMI), likely due to the initial acute-phase processes [34]. This finding suggests that proteolytic activity is essential for cardiovascular remodeling [35]. The role of MMP-9 as a vector for post-ischemic cardiac recovery and reconstruction has also been highlighted [36]. EMMPRIN is a key factor in triggering the migration and activation of MMPs [37].

VEGF mediates its effects on endothelial cells (EC) by activating VEGF receptor tyrosine kinases (VEGFR) [38]. VEGF-A activation of VEGFR-2, also known as human kinase insertion protein (KDR) receptor, is the major regulator of EC function. VEGF-activated VEGFR-2 modulates cell survival, migration, tube formation, and NO release [39]. Akt activates eNOS and increases NO production and release from EC, which is essential for promoting angiogenesis and remodeling of collateral vessels in response to ischemia [40]. It has been reported that androgens increase eNOS expression and NO production in HUVECs through AR-mediated genomic and non-genomic mechanisms [41]. They also promote angiogenesis, in part by activating the PI3-K/Akt-eNOS pathway in response to ischemic stress [42]. The results are in line with and extend our previous findings, namely that in MSCs, DHT promotes NO release and facilitates integration into cardiac tissue via ARs [26].

Angiogenin (ANG) plays an important role in angiogenesis by directly contributing to tissue remodeling, cell proliferation, and cell survival [43]. In endothelial cells, ANG translocates to the nucleus, binds to the promoter region, and stimulates transcription. In fact, the effects of angiogenic factors such as VEGF, FGF, and EGF can be blocked by using ANG inhibitors and inhibiting RNA [44].

We found that when EPCs indirectly interacted with heart tissue slices (indirect co-culture experiments), their ANG expression significantly increased. This finding suggests that ANG plays a central role in facilitating DHT-mediated EPC tissue repair and recovery, potentially working together with other factors.

Furthermore, the result of the direct co-culture experiment supports the notion that DHT is a powerful promoter of EPCs’ adhesion and integration into cardiac tissue. The two counting methods used confirmed that DHT treatment significantly increased the number of EPCs adhering to and residing within the heart slices, demonstrating the effect of the hormone beyond its effects on migration. Together, these results provide evidence of the important role of DHT in stabilizing EPCs to the site of injury. Moreover, the experiments in which androgen-receptor inhibitor was employed revealed that DHT affects the adhesion and integration of EPCs via an AR-mediated mechanism.

Our proteomic screening provides additional evidence for DHT-induced changes in EPCs. Per analyses conducted with Proteome Discoverer 1.4 and FunRich 3.1.3 software, we could assign the proteins showing significantly increased expression to specific biological processes and signaling pathways in DHT-stimulated EPCs, then represent these groupings in graphical form (Figure 9B).

Bioinformatic screening made it possible to group and highlight proteins based on their potential roles in cellular functions. The stimulated EPCs show upregulation of a significant number of proteins that support and regulate cell migration, angiogenesis, extracellular-matrix organization, metabolism of organonitrogen compounds, wound healing, and the response to wounding.

Mass spectrometry experiments, combined with bioinformatic tools and statistical analyses, allowed identification of upregulated (160) and downregulated (218) proteins.

The first 20 up-regulated proteins identified by Proteome Discoverer were analyzed based on their functions. Interestingly, the results confirm and extend the data showing that exposure of EPCs to DHT induces upregulation of Rho guanine nucleotide exchange factor 2 (ARHGEF2). Moreover, androgen deprivation restores ARHGEF2 to promote EPCs-mediated vasculogenesis and wound healing [45]. One can safely assume that binding of DHT to the androgen receptors induces upregulation of ARHGEF2 and the subsequent cellular processes. Among the proteins upregulated in stimulated EPCs, we detected OSR1. We presume that OSR1, also known as oxidative stress responsive kinase 1, is involved in the regulation of ion transport and responds to hormonal changes, including DHT (Table 1). It has been reported that WNK kinases, upstream regulators of OSR1, are involved in various physiological processes, including the regulation of blood pressure and ion homeostasis [46]. The dysregulation of WNK-SPAK/OSR1 signaling, influenced by factors such as DHT, has been linked to tumor growth, metastasis, and angiogenesis [47,48].

Identification of the top 20 most-downregulated proteins provides an overview of the potential enhanced function of DHT-exposed EPCs. As a result of downregulation of Rho GTPase-activating protein 1, the cells undergo changes in actin dynamics and cytoskeleton rearrangements, potentially enhancing EPCs’ migration and infiltration into ischemic tissues [12,49]. As shown in Table 2, in addition to these proteins, procollagen-lysine,2-oxoglutarate-5-dioxygenase 3 (PLOD3) and fibulin-1 are present. As has been reported, they could be responsible for creating a more flexible collagen matrix and loosening cell-matrix adhesions, allowing for easier mobilization and homing of EPCs to sites of vascular repair [50,51,52,53].

Taken together, our data on activated pathways and the supporting literature highlight the implication of DHT in various cellular and metabolic processes relevant to EPC-mediated tissue repair. Elucidation of the specific mechanisms and functional consequences of these changes holds promise for the development of novel therapeutic approaches. In some cardiovascular diseases, DHT-exposed EPCs could be valuable for vascular endothelium regeneration. However, care should be exercised when considering the pro-angiogenic effect of DHT. Our proof-of-concept experiments should be carefully adjusted and chosen according to the specifics of the disease.

Study limitation. The experiments performed in vitro on the effect of DHT on EPCs should be validated by in vivo experiments employing an adequate experimental model. In addition, the proteomic data should be confirmed and validated by a second assay.

## 4. Materials and Methods

### 4.1. EPCs Isolation and Characterization

EPCs were isolated and characterized from human cord blood (UCB) samples, as previously described [15]. Briefly, after collection, the blood was diluted 1:1 with phosphate-buffered saline (PBS), added to Histopaque 1.077 (Sigma-Aldrich, Burlington, MA, USA), and centrifuged (400× *g*). The mononuclear cells (MNCs) fraction was collected and washed three times with EBM-2 basal medium (Lonza, Basel, Switzerland) containing 10% fetal bovine serum (Gibco^®^, Life Technologies, Thermo Fisher Scientific, Waltham, MA, USA) and antibiotic-antimycotic solution (ZellShield, Minerva Biolabs GmbH, Berlin, Germany). Then, the MNCs were plated on collagen type I (rat tail; BD Biosciences, San Diego, CA, USA)-coated tissue-culture flasks and maintained in EGM-2 BulletKit (Lonza, Basel, Switzerland) medium under normoxic conditions. Cultured EPCs (at ~85% confluence) were further characterized by phase-contrast microscopy (Eclipse TE300, Nikon, Tokyo, Japan), for the presence of specific markers by flow cytometry (Gallios, BD Biosciences, San Diego, CA, USA), and the cells’ capacity to form tube-like structures in a three-dimensional (3D) system was assessed by Matrigel^®^ assay.

### 4.2. Flow Cytometry Analysis

Expression of surface molecules on EPCs was evaluated by flow cytometry (Gallios, BD Biosciences, San Diego, CA, USA). EPCs (1 × 10^5^ cells) were stained with fluorochrome-conjugated antibodies (phycoerythrin-PE, fluorescein isothiocyanate-FITC) specific for CD31, CD44, CD73, CD90, CD105, CD144 (R&D Systems, Minneapolis, MN, USA), and CD309 (MACS, Miltenyi Biotec, Bergisch-Gladbach, Germany).

EPCs were detached from the substrate using Accutase (Sigma Aldrich, Burlington, MA, USA), washed in PBS (1×) + 2% FBSCA solution and incubated for 30 min at 4 °C in dark with either PE or FITC-conjugated antibodies. As negative controls, cells were stained with isotype-specific IgG (IgG1, IgG2a/b, R&D Systems, Minneapolis, MN, USA). Flow cytometry data were analyzed by Summit 4.3 (Dako Santa Clara, CA, USA).

### 4.3. Characterization of Isolated EPCs: Formation of Tube-like Structures and the Uptake of Acetylated LDL

For the tube-formation assay, 50 μL Matrigel^®^ was added to a 48-well plate and allowed to solidify at 37 °C (30 min). Then, 5 × 10^4^ EPCs were resuspended in 100 μL culture medium and added to the Matrigel^®^ layer. After 24 h, the medium was removed and the formation of vascular tube structures was evaluated using an inverted microscope (Eclipse TS100; Nikon Minato, Tokyo, Japan) and a digital imaging camera (Digital SLR Camera D300; Nikon Minato, Tokyo, Japan).

For the ac-LDL uptake assay, EPCs were seeded at a density of 5 × 10^4^ cells/well in EGM2 medium in 24-well plates on glass slides. At ~80% confluence, the cells were exposed to serum-starved culture medium overnight in Dulbecco’s modified Iscove’s medium (IMDM, Sigma Aldrich, Burlington, MA, USA) supplemented with 2% serum substitute (Sigma Aldrich, Burlington, MA, USA). The medium was then replaced with IMDM supplemented with 100 μg/mL human ac-LDL. After 24 h, the cells on slides were stained with Nile Red (Sigma Aldrich, Burlington, MA, USA) and examined by fluorescence microscopy using an inverted microscope (Eclipse TS100; Nikon Minato, Tokyo, Japan) and a digital imaging camera (Digital SLR Camera D300; Nikon, Minato, Tokyo, Japan).

### 4.4. Assessment of the Effect of DHT on Cytotoxicity, Viability, and Proliferation of EPCs

Cytotoxicity, viability, and proliferation of DHT-treated EPCs were determined in real time using the xCELLigence system in conjunction with 16-View E-plate culture plates (ACEABiosciences, San Diego, CA, USA). Cells were seeded at a density of 5.000 cells/cm^2^ onto the E-plate and stimulated with DHT (AppliChem, Darmstadt, Germany) at various concentrations (1, 30, 100 and 1.000 nM) for the duration of the experiments. We chose from these conditions to use 30 nM DHT in further experiments because this concentration is similar to the concentration of circulating testosterone levels in human plasma and had no toxic effect on EPCs.

The impedance values of each well were automatically monitored for up to 96 **h**. Based on the cell impedance, the program monitored the value of the cell index (CI), which is directly correlated with the number of cells per well. This program allowed us to follow the evolution of cell proliferation rate over time in relation to the individual treatment. Experiments were performed in triplicate. The results obtained from proliferation tests were validated by the capacity of viable cells to incorporate MTT or 3-(4,5-dimethylthiazol-2-yl)-2,5-diphenyltetrazolium bromide, which is converted by cellular enzymes into a purple-colored insoluble form that can be spectrophotometrically quantified.

### 4.5. Real-Time Analysis of Migration of DHT-Stimulated EPCs Using an Improved Patented ‘Scratch Assay’ System

The system consisted of a modified polypropylene (PP) rectangle (…mm) inserted to the base of a well, part of a 16-well culture E-Plate and sealed with type I collagen. Thus, the base of the well was separated into two compartments. EPCs were added to the two compartments and maintained in an incubator until they reached confluence. Then, the PP rectangle was gently removed from the E-Plate, leaving behind an empty area similar to a scratch. After washing, the EGM2 culture media containing 30 nM DHT or 10 μM flutamide were added to the cells, followed after 4 h by 30 nM DHT. The migration of the cells to the empty area was continuously monitored for 84 h. The modified PP rectangle to be used for the xCELLigence E-Plates was the subject of a national patent registered at the Romanian Office of Trademarks and Inventions (No. 131463/2018).

### 4.6. Molecular Characterization of DHT Stimulated EPCs by qRT-PCR Technique

The quantitative expression of mRNA for AR, EMMPRIN, MMP-2, MMP-9, VEGFR-2, PlGF, and GAPDH (as an endogenous control) was evaluated. Total RNA was extracted using the PureLink™ RNA Mini Kit (Life Technologies™, Waltham, MA, USA). cDNA synthesis was performed at a concentration of 1 μg total RNA; cDNA samples were then amplified in triplicate using a qRT-PCR system (Applied Biosystems 7900HT Fast, Life Technologies™, Waltham, MA, USA) for 40 cycles (95 °C for 2 min, 95 °C for 5 s, 59 °C for 10 s, 72 °C for 15 s, plus one cycle of oligonucleotide dissociation sequences at 95 °C for 15 s and 60 °C for 15 s) using specific oligonucleotide sequences (EuroGenteck, Seraing, Belgium). qRT-PCR data were analyzed using SDS 2.4 Standalone (Applied Biosystems-Life Technologies™, Waltham, MA, USA). Oligonucleotide sequences for qRT-PCR were analyzed and selected using PerlPrimer software V1.1.21, and the nucleotide sequences of the primers are shown in Table 3.

### 4.7. Analysis of Proteins in DHT-Stimulated EPCs by Western Blot

Cultured EPCs were placed on ice, washed twice with cold PBS, lysed with Pierce Lysis Buffer and Halt Protease Inhibitor (Thermo Fisher Scientific, Waltham, MA, USA), and centrifuged (at 14,000× *g*, 10 min). After centrifugation, the supernatant was collected. Equal amounts of protein were separated on 10–12% polyacrylamide-SDS gels (Invitrogen, Waltham, MA, USA) under denatured conditions and transferred to Roti-PVDF 2.0 transfer membranes (Carl Roth, Karlsruhe, Germany). Nonspecific binding was blocked with 2% nonfat dry milk or BSA (2 h) in TBS (SantaCruz Biotechnology, Dallas, TX, USA). The membranes were incubated overnight (4 °C) with the following specific antibodies (1/1000 dilution): anti-human AR: PA5-16750 (Thermo Fisher Scientific, Waltham, MA, USA); MMP-2 anti-human: AV20016; MMP-9 anti-human: AV33090 (Sigma-Aldrich, Burlington, MA, USA); VEGFR-2 anti-human: ab39256 (Abcam, Cambridge, UK); PlGF anti-human: ab196666 (Abcam); and EMMPRIN anti-human: AF972 (R&D Systems, Minneapolis, MN, USA). Then, the membranes were washed with TBS and incubated with the corresponding HRP-conjugated secondary antibodies at a 1:200 dilution (1 h, room temperature): anti-rabbit HRP: AB6721 (Abcam); anti-mouse HRP: A9044 (Sigma-Aldrich, Burlington, MA, USA). The secondary antibodies were then detected by chemiluminescence (SuperSignal West Femto Maximum Sensitivity Substrate, Thermo Fisher Scientific, Waltham, MA, USA). Anti-human antibody to β-actin: sc-47778 (Santa Cruz Biotechnology, Dallas, TX, USA) was used as a control. Blots were scanned using an imaging system (ImageQuant LAS-3000, Fujitsu Life Sciences, Minato-ku, Tokyo, Japan). Target signals were normalized to β-actin and analyzed semi-quantitatively using ImageJ Version 1.54f (NIH) image-processing software.

### 4.8. Preparation of Murine Cardiac Tissue Slices to Study, the Interaction with EPCs Ex Vivo

All animal experiments were performed in accordance with the current EU legislation establishing the legal framework for the use of animals in scientific research after approval by the Ethics Committee of the “Nicolae Simionescu” Institute of Cell Biology and Pathology. Harvested hearts were processed as previously reported [54] and embedded in 4% low-melting-point agarose (Roth, Karlsruhe, Germany), and the tissue was cut into 250-μm-thick sections along the short axis with a vibratom (Leica VT1000S, Leica Microsystems, Wetzlar, Germany). Murine ventricular sections were kept in Tyrode’s solution supplemented with 0.9 mmol/L CaCl_2_ and bubbled with O_2_ for 30 min, then transferred to DMEM supplemented with 20% FBS and 1% antibiotic/antifungal and placed in an incubator for 1 h at 37 °C and 5% CO_2_.

### 4.9. Real-Time Assessment of EPCs’ Chemotaxis to Heart Tissue Slices Using the xCELLigence System

The xCELLigence system has the advantage of measuring cell migration accurately and in real time by using the CIM-plates adapters [55]. Mouse ventricular slices were placed in the lower chamber, and 4 × 10^4^ EPCs were added to the upper chamber. The cells were exposed to one of three conditions: previously stimulated with DHT, non-stimulated, or exposed to the vehicle (as controls). DMEM (1×) with GlutaMAX and 10% FBSCA and 1% antifungal/antibiotic solution (Sigma-Aldrich, Burlington, MA, USA) was used in both chambers. CIM-plates were kept in an incubator under normoxic conditions (as above). To evaluate the role of AR in the migration process, EPCs were treated with 10 µM flutamide (45 min) prior to 30 nM DHT stimulation. Cell migration was monitored for 20 h. Determinations were performed in four (4) technical replicates, and experiments were repeated at least three times.

### 4.10. Luminex Technique Analysis of EPCs-Secreted Proteins in Culture Media after Their Indirect Contact with Cardiac Tissue

For these experiments, 24-well Transwell culture plates with inserts containing a 0.4 μm-pore polycarbonate basement membrane treated for tissue culture (Corning- Corning NY, USA) were used. Cardiac tissue slices were placed in the lower chamber, and 4 × 10^4^ EPCs that had been stimulated with DHT or that had not been previously stimulated with DHT were cultured in the upper chamber. Both cells and tissue sections were maintained in DMEM medium (1×) with GlutaMAX containing 10% FBSCA and 1% antibiotic/antimycotic solution and cultured in an incubator under normoxic conditions. For each experimental condition, 1.5 mL of conditioned medium was collected from the lower chamber at 24- and 48-h intervals to determine the levels of MMP-2 and MMP-9 proteins using ELISA assays. Similar experimental conditions were used to evaluate the expression of angiogenin and VEGF proteins using the Luminex Human Angiogenic Panel A (BD Bioscience, San Diego, CA, USA) technique according to the manufacturer’s protocol.

### 4.11. Determination of the Adhesion/Integration Capacity of EPCs Placed in Direct Contact with Cardiac Tissue Sections by Immunohistochemistry

In a 48-well treated tissue-culture plate (Corning, Corning, NY, USA), 250 μm murine ventricular slices were attached to the bottom of the culture plate with 5 μL type 1 collagen (as above), over which DMEM medium supplemented was added. Next, 4 × 10^4^ EPCs treated (96 h) or untreated with DHT were added on top of the cardiac tissue, and after 72 h of co-culture, the ventricular sections were washed twice with PBS. The evaluation of the number of migrated EPCs was done by immunochemistry (to identify human EPC nuclei) or by DNA quantification. For immunohistochemistry, heart fragments were fixed in 4% paraformaldehyde overnight and embedded in paraffin. Microtome sections (DTK-2000, Dosaka, Kyoto, Japan) of ~2 μm were placed on microscope slides, deparaffinized, and rehydrated through washing with successive concentrations of xylene and ethanol (100, 95, 70, and 50%). To facilitate antibody binding, the sections were placed (3 min) in sodium citrate buffer in a pressure vessel.

Integration of human EPCs into murine ventricular sections was determined upon exposure of sections to the primary anti-human nuclei antibody (1:100, Sigma-Aldrich, Burlington, MA, USA), which was followed by exposure to the HRP-conjugated anti-mouse secondary antibody (1:1000, A9044, Sigma-Aldrich, Burlington, MA, USA); as substrate for the detection system, we employed 3,3′-diaminobenzidine (DAB) (Sigma-Aldrich, Burlington, MA, USA).

### 4.12. qRT-PCR Quantification of Human DNA for the Assessment of the Adhesion/Integration of Human EPCs upon Contact with Cardiac Tissue Sections

For these experiments, we used Real 7900 HT PCR (as above), TaqMan Universal PCR Master Mix-4304437, Human GAPDH-Hs03929097_g1, and a 384-well PCR plate (Thermo Fisher Scientific, Waltham, MA, USA). Human EPCs (4 × 10^4^), untreated or DHT-treated (30 nM), were incubated as described for immunocytochemistry experiments. After 48 h, DNA was isolated using the PureLink Genomic DNA Isolation Kit (Life Technologies, Waltham, MA, USA) and quantified by the NanoDrop ND-1000 UV-VIS spectrophotometer. To accurately determine the relative cellularity, a calibration curve was established by serially diluting standardized DNA extracted from a reference sample of 4 × 10^4^ EPCs to a final dilution of 2500 equivalent cells [56]. In addition, for the design of oligonucleotide sequences, we compared the human genome to the mouse genome using the Genome ARTIST V1 software [57]. Experimental conditions for PCR were chosen according to the manufacturer’s specifications, and all experiments were performed in triplicate.

### 4.13. Analysis of EPCs Protein Profiles Stimulated or Not with DHT by Nano Liquid Chromatography- Mass Spectrometry (LC-MS)

Urea, sodium deoxycholate (DOC), Tris-HCl, dithiothreitol, iodoacetamide, N-acetyl-1-cysteine, ammonium bicarbonate, and MS solvents were purchased from Sigma-Aldrich (Burlington, MA, USA). Trypsin Gold was obtained from Promega, and Roche protease inhibitors and C18 solid phase extraction columns were acquired from Waters (Milford, MA, USA). The ADV-01A kit from Tebu-Bio (Le Perray-en-Yvelines, France) was used for protein quantification. EPCs were washed twice with PBS and centrifuged, and the pellet was solubilized in a denaturing buffer containing 8 μm urea, 1% DOC, 0.1% Tris-HCl (pH 8.8), and protease inhibitors by sonication for 30 s. on ice (UP50H from Hielscher, Teltow, Germany). Protein purification was carried out as previously described [58]. LC-MS analysis was performed using an EASY n-LC II system (Thermo Scientific, Waltham, MA, USA) coupled to the LTQ Orbitrap Velos Pro mass spectrometer (Thermo Scientific, Waltham, MA, USA), operating in 60k data-dependent Top 12 Data Dependent analysis [59]. Three biological replicates, each in technical triplicate, were processed and analyzed for peptide separation and mass-spectra acquisition. Bioinformatics analysis, including protein inference, relative quantification, and determination of statistically over-represented Kyoto Encyclopedia of Genes and Genomes (KEGG) signaling pathways, was performed as previously described [60].

## 5. Statistical Analysis

Medians, standard deviations, and *p*-values were calculated using Microsoft Excel Version Plus 2016 (Microsoft, Redmond, WA, USA). Standard-curve equations, coefficients of determination (R2), and efficiencies (E) were obtained using LightCycler^®^ 480 Software 1.5. R^2^ is the regression coefficient. PCR efficiency (E) is obtained from E = 10^−1^/s^−1^, where *s* is the slope of the regression line between Cp values and log cells [61]. Cp values were plotted versus the log (number of cells) of six serial dilutions of DNA obtained from 4 × 10^4^ cells. Medians were compared by Student’s *t*-test using Microsoft Excel. Results were considered significant if *p* ≤ 0.05.

## 6. Conclusions

DHT stimulates proliferation, viability, migration, and angiogenesis of human umbilical-blood-derived EPCs through an androgen-receptor-dependent mechanism. At the molecular level, these attributes manifest in the regulated expression of genes and proteins connected to cell migration and in the enhanced synthesis of key pro-angiogenic factors. Furthermore, DHT improves the integration of EPCs into the cardiac tissue in an androgen-receptor-dependent manner, indicating the potential role of hormone preconditioning in promoting tissue regeneration. Mass spectrometry analysis strengthened the evidence that hormone-modulated proteins are implicated in cellular processes such as migration, angiogenesis, metabolic processes, wound healing, and response to wounding. The results highlight the potential of DHT to augment the native properties of EPCs and the potential of stimulated cells to become a better therapeutic agent for cardiovascular regeneration and repair.

## Figures and Tables

**Figure 1 ijms-25-04862-f001:**
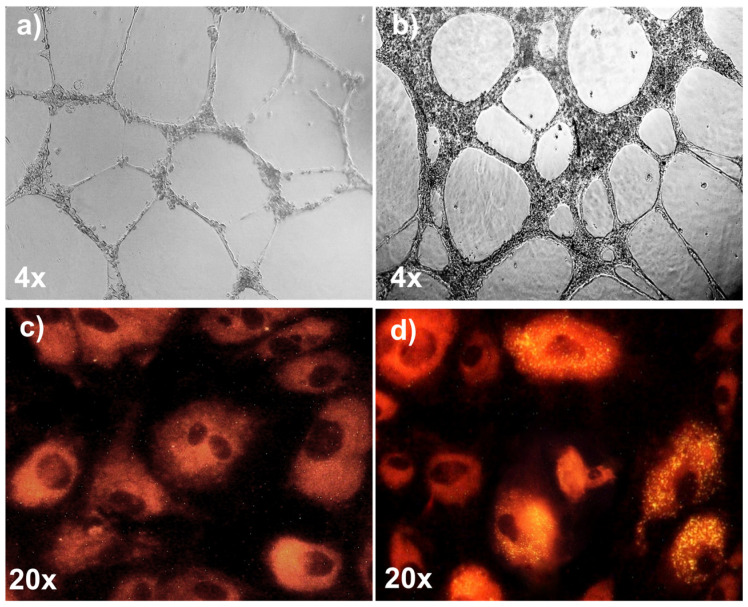
Functional characterization of EPCs isolated from umbilical-cord blood. (**a**) Isolated EPCs were cultured on Matrigel^®^ substrate. At 4 h (**a**) and at 24 h (**b**) after seeding, the cells formed robust capillary-like tubes. At 24 h after seeding, compared to controls (**c**) isolated EPCs avidly took up acetylated LDL (yellow dots), as assessed by fluorescence microscopy (**d**).

**Figure 2 ijms-25-04862-f002:**
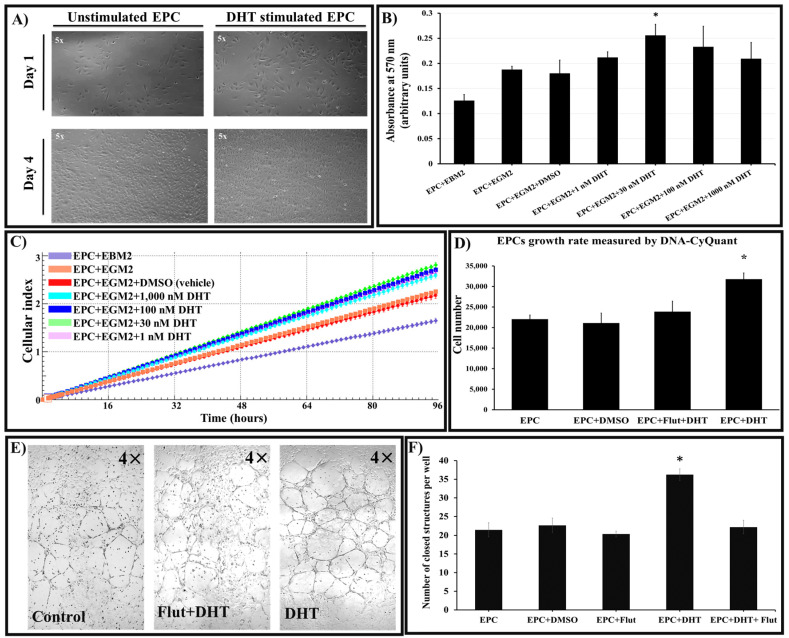
Exposure of EPCs to DHT does not alter cell morphology and significantly increases cell proliferation and function by an androgen-receptor-dependent mechanism. (**A**) Morphological evaluation of EPCs at days 1 and 4 after seeding showed that 30 nM DHT treatment does not alter the characteristic spindle-shaped morphology indicative of their endothelial identity. (**B**) EPCs were cultured in EGM2 medium supplemented with varying concentrations of DHT (1, 30, 100, and 1000 nM) or vehicle. MTT assay shows that after 96 h of stimulation, the presence of 30 nM DHT significantly increases cell proliferation compared to control (EPC+EGM2). (**C**) Analysis on an xCELLigence real-time cell analyzer shows that at all concentrations used, DHT induces in EPCs (cultured as above) a similar mitogenic effect and an increase in cell viability (compared to controls). (**D**) Assessment of EPC proliferation by quantification of DNA content using the CyQuant assay. The DHT treatment (30 nM) results in a ~30% increase in EPCs’ DNA content compared to controls (EPC and EPC+DMSO). Given prior treatment of cells with 10 µM flutamide (Flut), an androgen-receptor antagonist, DHT significantly reduces EPC proliferation, indicating a role for androgen receptors in proliferation. (**E**) Compared to controls, DHT-exposed EPCs exhibit increased formation of tube-like structures in Matrigel-coated wells. Note the inhibitory effect of flutamide (Flut+DHT). (**F**) Morphometric analyses showing that DHT treatment significantly enhances (50%) the formation of closed capillary-like structures compared to controls (EPC, EPC+DMSO); flutamide significantly reduces this property, indicating a role for AR in EPCs’ formation of tube-like structures. n = 3. * *p* value < 0.05. EGM2 = endothelial growth media 2; EBM2 = endothelial basal medium 2; DMSO = dimethyl sulfoxide.

**Figure 3 ijms-25-04862-f003:**
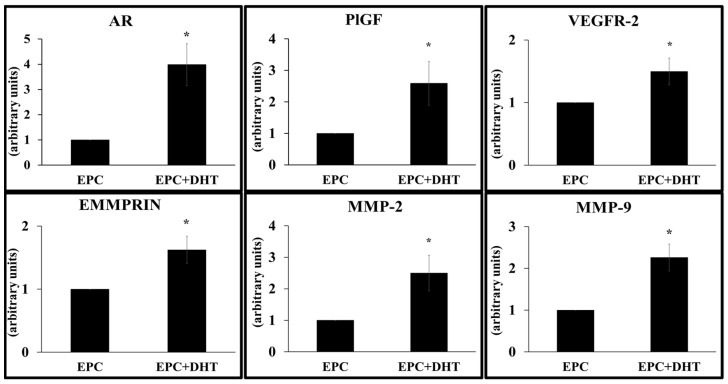
Quantification of the gene expression in DHT-stimulated EPCs by qRT-PCR. EPCs exposure to 30 nM DHT (EPC+DHT) increases gene expression of androgen receptors (AR), EMMPRIN, MMP-2, MMP-9 as well as VEGFR-2 and PlGF. Data are expressed as means (±) SEM of each gene relative to GAPDH and normalized to an arbitrary value of 1. n = 3. * *p* value < 0.05.

**Figure 4 ijms-25-04862-f004:**
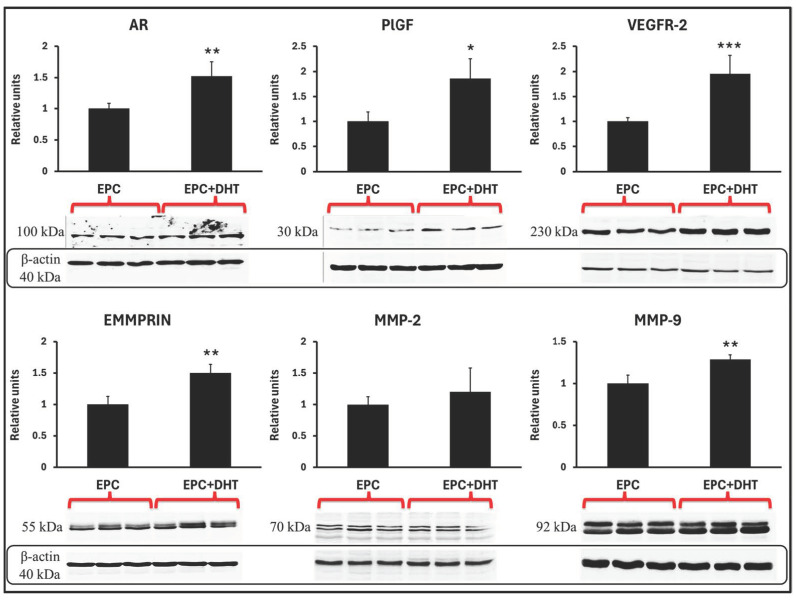
Western blot analysis of proteins expressed by EPCs stimulated with DHT (EPC+DHT) shows significant increases in the levels of the proteins AR, PlGF, EMMPRIN, VEGFR-2, and MMP-9. Compared to controls (EPCs), DHT induces a significant increase in the expression of all proteins except for the MMP-2 protein. All results were normalized to levels of β-actin expression; * *p* value < 0.05, ** *p* value < 0.001, *** *p* value < 0.0001, n = 3.

**Figure 5 ijms-25-04862-f005:**
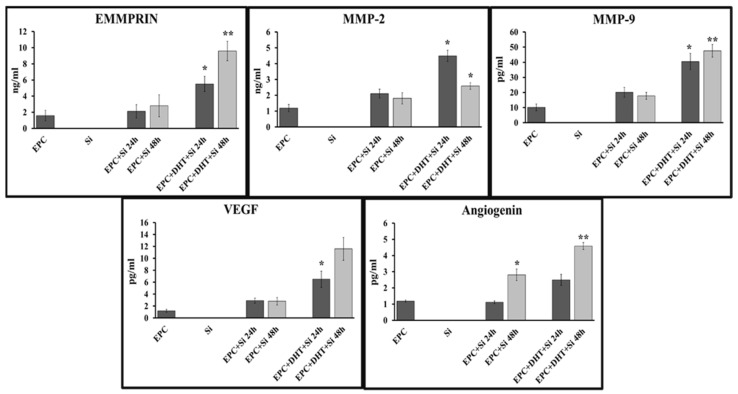
DHT-exposed EPCs were seeded in the upper chamber of a Transwell plate, and the heart tissue slices (Si) were placed in the lower chamber. After 24 and 48 h, the medium was collected and centrifuged and the proteins in the supernatant were quantified using Human Angiogenesis Kit Panel A—Luminex assay. Note that the secretion of the proteins EMMPRIN, MMP-9, angiogenin, and VEGF is significantly increased in the media collected from DHT-stimulated EPCs (EPC+DHT+Si 24 h/48 h) upon indirect contact with heart fragments compared to controls (EPC+Si 24 h/48 h). * *p* < 0.05; ** *p* < 0.001. n = 3.

**Figure 6 ijms-25-04862-f006:**
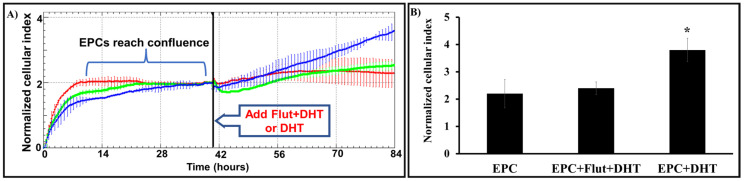
DHT increases the migration capacity of EPCs by a mechanism dependent on androgen receptors. The cells plated on our E-Plate patented device reached confluence, and after ~40 h, DHT or flutamide+DHT were added to the wells. Real-time measurements of the cell migration were conducted via the xCELLigence system. The representative histogram (**A**) and the cumulative normalized cellular index (**B**) show that compared to that of non-stimulated cells (red), the speed of migration of DHT-stimulated EPCs (blue) was significantly higher. Exposure of cells to flutamide (green) prior to exposure to DHT (EPC+Flut+DHT) reduces the EPCs’ migration by ~50%. * Value *p* < 0.05, n = 3.

**Figure 7 ijms-25-04862-f007:**
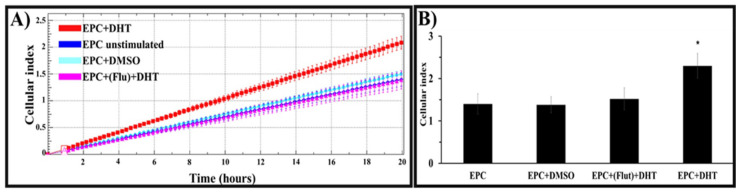
Effect of DHT on EPCs’ migration towards ventricular slices, as quantified by real-time measurements of cellular impedance using xCELLigence’s system (CIM-Plate). As shown by the representative histogram (**A**) and the cumulative cellular index (**B**), after 20 h of measurements, the cell index, indicating the number of migrating cells, is significantly higher for DHT-treated EPCs (red) compared to controls (untreated cells (blue) and vehicle (DMSO)-treated cells (cyan)). Exposure of cells to flutamide (pink) significantly reduces the effect of the hormone. Experiments were performed in quadruplicate, n = 3; * *p* value < 0.05.

**Figure 8 ijms-25-04862-f008:**
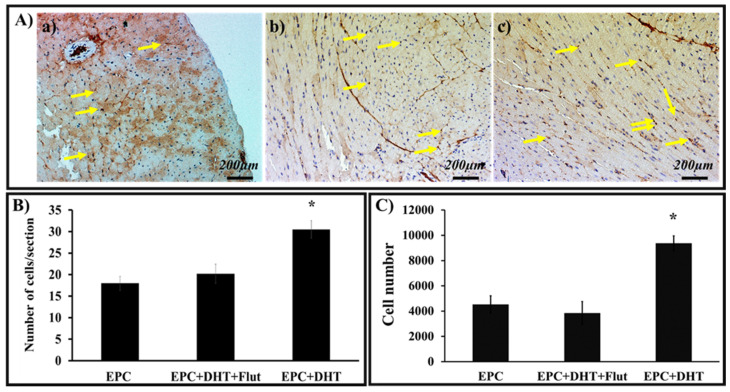
Adhesion and integration of DHT-treated EPCs in ventricular tissue, as detected by immunohistochemistry and quantified by ImageJ and qRT-PCR. EPCs exposed or not exposed to 30 nM DHT in the presence or absence of flutamide (Flut) were co-cultured with heart fragments, washed, and embedded in paraffin. Heart sections were incubated successively with anti-human nuclear primary antibody and HRP-labelled secondary antibody. After the peroxidation reaction, the human EPCs nuclei were stained brown (yellow arrows), while the mouse cell nuclei were blue. (**A**) Compared to controls (**a**), the number of brown-stained human nuclei was higher in heart fragments co-cultured with stimulated EPCs (**b**); Flut considerably reduces the integration of EPCs (**c**). (**B**) Digital counting of stained heart sections confirmed the observations that the number of human cell nuclei, indicating the integration of DHT-exposed EPCs (EPC+DHT), is significantly higher compared to controls (EPC) and that the Flut-induced AR blockade reduces the number of adherent cells (EPC+DHT+Flut). For each section, four different fields were analyzed (n = 3). (**C**) Quantification of the number of adhered/integrated EPCs co-cultured with heart sections, as assessed by human DNA analysis (qRT-PCR). n = 3. * *p* value < 0.05 compared to untreated cells.

**Figure 9 ijms-25-04862-f009:**
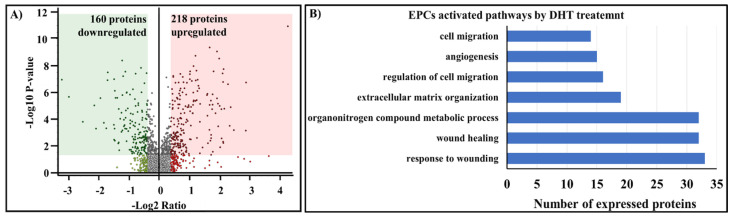
Nano-liquid chromatography–mass spectrometric analysis of DHT-treated EPCs and non-treated cells. (**A**) Qualitative analysis by Proteome Discoverer identified 2521 proteins, of which 160 proteins are downregulated and 218 proteins are upregulated. The volcano plot highlights proteins that have undergone significant changes in abundance between the two groups, as determined by spectral abundance of the EPCs proteins. The green and red rectangles in the scatterplot represent the threshold values for the log_2_ normalized ratio (−0.33) and −log_10_ *p* value (0.05), respectively, indicating biological alteration corroborated with statistical significance. (**B**) Gene ontology analysis of differentially expressed proteins from EPCs. DHT-exposed EPCs exhibit deregulation of proteins involved in tissue regeneration mechanisms. The x-axis represents the number of differentially expressed proteins, and the y-axis lists biological processes extracted from the FunRich database (FunRich version 3.1.3) associated with the proteins showing statistically significant evidence of regulation.

**Figure 10 ijms-25-04862-f010:**
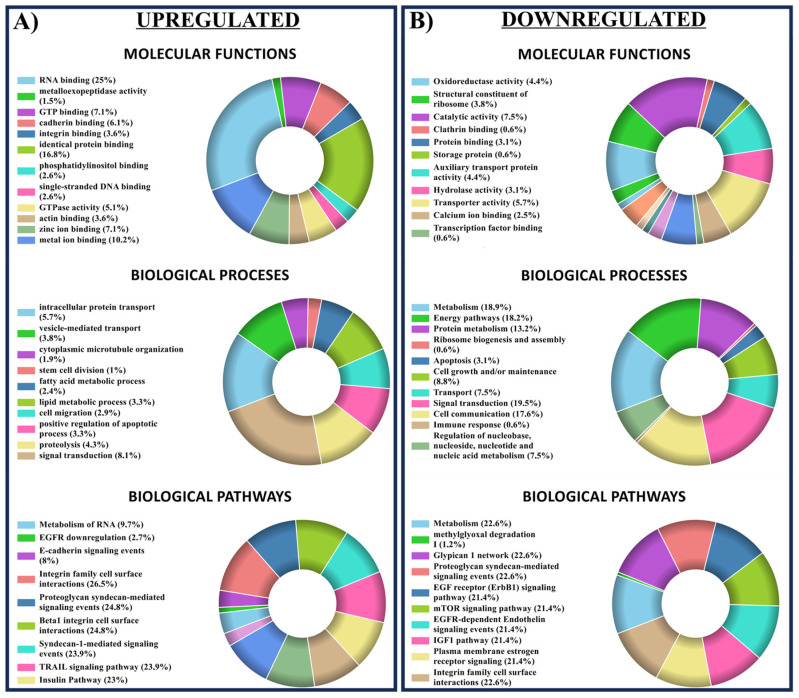
Bioinformatic analyses (using FunRich 3.1.3—Uniprot database) cluster the DHT-stimulated EPC proteins that are upregulated (**A**) and downregulated (**B**) into categories related to their molecular functions, biological processes, and signaling pathways. The proteins participating in different cellular processes, identified by bioinformatic tools, correspond to a false-discovery-rate-corrected *p*-value of <0.05. The EPCs proteins with different associations are represented as percentages of the total cellular protein components.

**Table 1 ijms-25-04862-t001:** Top twenty most-upregulated proteins in DHT-stimulated EPCs, as identified using Proteome Discoverer Software-Version 1.4.

Accession	Description	Abundance Ratio: (DHT)/(Control)	*p*-Value: (DHT)/ (Control)
Q9C0H2	Protein tweety homolog 3	19.16	1.18 × 10^−8^
Q92974	Rho guanine nucleotide exchange factor 2	7.374	1.31 × 10^−5^
O95747	Serine/threonine-protein kinase OSR1	7.315	8.2 × 10^−3^
O15230	Laminin subunit alpha-5	5.569	7.74 × 10^−3^
P30622	Isoform 2 of CAP-Gly domain-containing linker protein 1	5.16	3.32 × 10^−4^
P19388	DNA-directed RNA polymerases I, II, and III subunit RPABC1	5.149	2.35 × 10^−9^
Q09161	Nuclear cap-binding protein subunit 1	5.049	4.11 × 10^−3^
P43307	Translocon-associated protein subunit alpha	4.853	1.64 × 10^−5^
O00468	Agrin	4.48	4.07 × 10^−4^
Q8NE71	ATP-binding cassette sub-family F member 1	4.405	3.07 × 10^−6^
P33316	Deoxyuridine 5’-triphosphate nucleotidohydrolase, mitochondrial	4.364	4.11 × 10^−2^
P03956	Interstitial collagenase	4.346	3.47 × 10^−2^
Q9BWD1	Acetyl-CoA acetyltransferase, cytosolic	4.339	1.30 × 10^−5^
Q96AP7	Endothelial cell-selective adhesion molecule	4.304	5.05 × 10^−6^
O95084	Serine protease 23	4.213	2.18 × 10^−2^
P06454	Prothymosin alpha	4.184	4.19 × 10^−4^
Q5SWX8	Protein odr-4 homolog	4.075	7.2 × 10^−2^
P26885	Peptidyl-prolyl cis-trans isomerase FKBP2	4.018	2.83 × 10^−4^
P53680	AP-2 complex subunit sigma	3.973	9.97 × 10^−7^

**Table 2 ijms-25-04862-t002:** Top twenty most-downregulated proteins in EPCs exposed to DHT pre-treatment, as identified using Proteome Discoverer Software-Version 1.4.

Accession	Description	Abundance Ratio: (DHT)/ (Control)	*p*-Value: (DHT)/ (Control)
P05106	Integrin beta-3	0.541	1.58 × 10^−1^
Q9C0C2	182 kDa tankyrase-1-binding protein	0.669	5.32 × 10^−5^
Q9P0V3	SH3 domain-binding protein 4	0.363	4.30 × 10^−5^
O60568	Procollagen-lysine,2-oxoglutarate 5-dioxygenase 3	0.751	4.98 × 10^−6^
O00541	Pescadillo homolog	0.506	9.65 × 10^−2^
P17252	Protein kinase C alpha type	0.639	2.03 × 10^−2^
Q8IY17	Neuropathy target esterase	0.446	1.73 × 10^−3^
P15374	Ubiquitin carboxyl-terminal hydrolase isozyme L3	0.464	1.44 × 10^−3^
P11310	Medium-chain specific acyl-CoA dehydrogenase, mitochondrial	0.356	2.66 × 10^−2^
P82909	28S ribosomal protein S36, mitochondrial	0.712	9.52 × 10^−3^
P23142	Fibulin-1	0.401	1.05 × 10^−1^
Q07960	rho GTPase-activating protein 1	0.612	6.27 × 10^−4^
Q8N126	Cell adhesion molecule 3	0.587	1.15 × 10^−1^
O94875	Isoform 11 of Sorbin & SH3 domain-containing protein 2	0.428	1.15 × 10^−6^
P51808	Dynein light chain Tctex-type 3	0.306	1.06 × 10^−5^
Q969X5	Endoplasmic reticulum-Golgi intermediate compartment protein 1	0.515	1.85 × 10^−3^
P51858	hepatoma-derived growth factor	0.746	3.19 × 10^−2^
Q13492	Phosphatidylinositol-binding clathrin assembly protein	0.762	1.54 × 10^−1^
Q16740	ATP-dependent Clp protease proteolytic subunit, mitochondrial	0.734	1.61 × 10^−3^

**Table 3 ijms-25-04862-t003:** List of human specific primers identified using PerlPrimer Software Version 1.1.21.

Primer (Human)	Forward Primer (5′-3′)	Reverse Primer (5′-3′)
GAPDH	GTTTCTATAAATTGAGCCCGCAG	CGACCAAATCCGTTGACTCC
Androgen receptor (AR)	ATCCTTCACCAATGTCAACTCC	CCACTGGAATAATGCTGAAGAG
MMP-2	ACTACAACTTCTTCCCTCGCA	GGCATCATCCACTGTCTCTG
MMP-9	GCCACTACTGTGCCTTTGAG	CAGAGAATCGCCAGTACTTCC
EMMPRIN	CTCACCTGCTCCTTGAATGAC	GAGTCCACCTTGAACTCCGT
VEGFR-2	AAGTAATCCCAGATGACAACCA	CCTTCAGATGCCACAGACTC

## Data Availability

The data presented in this study are available on request from the corresponding author (MS).

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
