# Peer review of "Dihydrotestosterone Augments the Angiogenic and Migratory Potential of Human Endothelial Progenitor Cells by an Androgen Receptor-Dependent Mechanism"

_ijms, 2024, doi:10.3390/ijms25094862_

Round 1

Reviewer 1 Report

Comments and Suggestions for Authors

The research article entitled "Dihydrotestosterone augments the angiogenic and migratory potential of human endothelial progenitor cells by an androgen-dependent mechanism" submitted by Pope and colleagues addresses an important issue in cardiovascular physiology/biology. The submitted work also has the potential for the development of novel therapeutic targets in the future. My specific comments about the manuscript are as follows:

1. In the title the phrase 'androgen-dependant mechanism' seems redundant as the said mechanism is initiated by a potent androgenic hormone, instead the 'dose-dependant mechanism' can be used if the authors feel the need to hint towards the mechanism in the title. 

1. Abstract section line 14: There is an extra space before the start of the sentence.

2. Abstract section line 18: The authors should elaborate very briefly on the experimental setup before mentioning the results.

3. Introduction section: The overall format of the introduction should be amended. Instead of using multiple paragraphs consisting of 2-3 lines, the authors should format the introduction into 2 main paragraphs followed by a third smaller paragraph. Authors should also focus more on different parameters in the introduction that they are assessing in their experiments. 

4. Introduction section line 54: The first sentence should be rephrased.

5. Introduction section line 55: the line should read as "However there are also some conflicting reports which don't support this notion".

6. Introduction section line 58: instead of using the word 'assume', authors should rephrase it as "it is speculated that the mechanism....".  

7. Introduction section lines 76-79: The authors mention the result of the study in the introduction which is not advised. Instead, the authors should use this last paragraph to explain their experimental design and the specific aspects that they ought to explore in their experiments.

8. Result section line 108: the 'assay' should be replaced with 'assess'.

9. Result section line 112: The sentence "For the following experiments we choose to use 30nM......" is not followed by experiments, and instead the reader is faced with the end of the paragraph and the next paragraph starts with a new aspect e.g. Androgen receptor. Secondly, if the authors insist on explaining the reason for using 30nM conc of DHT, they should explain it in the "materials and methods" section, and not in the result section.

10. Result section lines 115-116: The start of the paragraph is incorrect. the word 'Notably' should be backed by a previous sentence mentioning multiple aspects, out of which, the authors focus on one specific aspect and highlight it in the next sentence by starting the sentence with the word 'notably'. Secondly, the proliferation of EPCs was found to be contingent (the word 'contingent' should be replaced by 'dependant') upon the androgen receptor what? the number of androgen receptors? the activation of the androgen receptors? availability of the unbound receptors?

11. Figure 2. Legend. Line 126: B) should also be in bold.

12. Discussion section: The whole discussion section contains less than 20 references, while the authors have presented a monumental amount of data to which the discussion section in its current form doesn't do any justice. 

Author Response

We thank the reviewer for excellent comments and constructive criticism of our manuscript. All the comments were well taken, and we believe that they improved our revised manuscript.

Please find below a point-by-point answer to reviewer comments:

Comment 1: In the title the phrase 'androgen-dependant mechanism' seems redundant as the said mechanism is initiated by a potent androgenic hormone, instead the 'dose-dependant mechanism' can be used if the authors feel the need to hint towards the mechanism in the title.

Answer to Comment: We appreciate the reviewer’s suggestion. We agree that the term ‘androgen-dependent mechanism’ might seem redundant. However, we used this term to emphasize the specific role of androgens in the mechanism under study (on human progenitor cells).

Comment 2: Abstract section line 14: There is an extra space before the start of the sentence.

Answer: The extra space was deleted.

Comment 3: Abstract section line 18: The authors should elaborate very briefly on the experimental setup before mentioning the results.

Answer: There reviewer comment was well taken, and we added in the revised manuscript a brief sentence about experimental setup (page 1, lines 16-18).

Comment 4: Introduction section: The overall format of the introduction should be amended. Instead of using multiple paragraphs consisting of 2-3 lines, the authors should format the introduction into 2 main paragraphs followed by a third smaller paragraph. Authors should also focus more on different parameters in the introduction that they are assessing in their experiments.

Answer: We highly appreciate the reviewer’s feedback on the structure of the Introduction. We have revised the entire introduction. In the revised manuscript there are two main paragraphs, followed by a third smaller paragraph focusing on the parameters assessed in our experiments.

Comment 5: Introduction section line 54: The first sentence should be rephrased.

Answer: For clarity, the first sentence was rephrased (page 2, lines 47-50).

Comment 6: Introduction section line 55: the line should read as "However there are also some conflicting reports which don't support this notion".

Answer: We rephrased as suggested the sentences (new page 2, line 49).

Comment 7: Introduction section line 58: instead of using the word 'assume', authors should rephrase it as "it is speculated that the mechanism....". 

Answer: We rephrased as suggested the sentences (new page 2, line 51)

Comment 8: Introduction section lines 76-79: The authors mention the result of the study in the introduction which is not advised. Instead, the authors should use this last paragraph to explain their experimental design and the specific aspects that they ought to explore in their experiments.

Answer to Comment: The reviewer’s comment was well accepted, and we revised the introductory section as suggested (page 2 lines 63-67).

Comment 9: Result section line 108: the 'assay' should be replaced with 'assess'.

Answer:  Thank you, the word was removed and we rephrase the sentence in order to increase its accuracy.

Comment 10: Result section line 112: The sentence "For the following experiments we choose to use 30nM......" is not followed by experiments, and instead the reader is faced with the end of the paragraph and the next paragraph starts with a new aspect e.g. Androgen receptor. Secondly, if the authors insist on explaining the reason for using 30nM conc of DHT, they should explain it in the "materials and methods" section, and not in the result section.

Answer: We appreciate the reviewer’s feedback. As suggested, we explained the reason for using 30 nM DHT and the sentence was placed to the appropriate section. The entire sentence was rephrased for clarity (page 3 lines 96-98).

Comment 11: Result section lines 115-116: The start of the paragraph is incorrect. the word 'Notably' should be backed by a previous sentence mentioning multiple aspects, out of which, the authors focus on one specific aspect and highlight it in the next sentence by starting the sentence with the word 'notably'. Secondly, the proliferation of EPCs was found to be contingent (the word 'contingent' should be replaced by 'dependant') upon the androgen receptor what? the number of androgen receptors? the activation of the androgen receptors? availability of the unbound receptors?

Answer: Thank you for the comments. We have revised the paragraphs, remove the ‘Notably’ and replaced the word ‘contingent’ with ‘dependent’ and clarified that the proliferation of EPCs is dependent on the ‘availability of AR’ (page 3, lines 99-100).

Comment 12: Figure 2. Legend. Line 126: B) should also be in bold.

Answer: We have corrected it and made ‘B)’ bold in the legend of Figure 2 at page 4, line 111.

Comment 13: Discussion section: The whole discussion section contains less than 20 references, while the authors have presented a monumental amount of data to which the discussion section in its current form doesn't do any justice.

Answer: We agree that our discussion section should reflect the breadth and depth of the current knowledge. Therefore, in the revised manuscript we added to the Discussion section 18 new references to acknowledge the work of other scientists in the field.

We thank to the reviewer for the time and attention paid to our manuscript. The comments were very helpful, we revised the entire manuscript, and we hope that in the revised form will meet the criteria of IJMS.

The authors

Reviewer 2 Report

Comments and Suggestions for Authors

In this study, Popa et al. investigated the role of dihydrotestosterone (DHT) in cell function of endothelial progenitor cells (EPCs). The data showed that DHT increases EPCs proliferation, migration, angiogenesis, and enhances the secretion of angiogenic factors. This research is of significant importance, but there are still some concerns that need to be addressed. In particular, adding some animal experiment validation would greatly enhance the completeness of the data.

Comments:

1)     The study only employs one cell line. It is suggested to use another type of endothelial cell line to validate the key findings, for example, primary human umbilical vein endothelial cells (HUVECs) and human iPSC-derived endothelial cells. Would DHT also improve their function?

2)     Similarly, it's suggested to validate the impact of DHT on angiogenesis through in vivo experiments, such as the Matrigel plug assay.

3)     It's recommended to add a cell scratch assay to demonstrate the effect of DHT on EPCs.

4)     In Figure 4, please clearly indicate which bands belong to the EPC group and which to the EPC+DHT group. Additionally, the molecular weight information is unclear; please replace it with an electronic version instead of a handwritten format.

5)     In the statistical results of the Western blot experiment, why was the EPC control group not normalized to 1?

Comments on the Quality of English Language

 Minor editing is required.

Author Response

 We thank the reviewer for the appreciation paid to our paper, for the comments and constructive criticism. All the comments were well taken, and they helped us to improve our revised manuscript. Please find below a point-by-point answer to the comments.

Comments: 

  • The study only employs one cell line. It is suggested to use another type of endothelial cell line to validate the key findings, for example, primary human umbilical vein endothelial cells (HUVECs) and human iPSC-derived endothelial cells. Would DHT also improve their function?

Answer: While human umbilical vein endothelial cells (HUVECs) and human iPSC-derived endothelial cells are valuable tools for in vitro studies, we specifically chose human umbilical blood-derived EPCs because they are considered a promising candidate for therapeutic applications in cardiovascular regeneration. HUVECs and iPSC-derived endothelial cells may not fully represent the characteristics and regenerative potential of human EPCs; in addition, they have limited chances to be used as a therapeutic option in vivo; one is a venous type of EC, and iPSCs are not indicated for regenerative therapy. Therefore, focusing on EPCs strengthens the translational relevance of our findings for potential therapeutic development. We agree that in vitro studies require in vivo validation, and we have acknowledged this as a limitation of the current research (end of Discussion section).

  • Similarly, it's suggested to validate the impact of DHT on angiogenesis through in vivo experiments, such as the Matrigel plug assay.

Answer: This is a good suggestion and fits well with our plans to validate our results in vivo. In a previous paper, we reported the effect of DHT on mesenchymal stem cells in vivo using the Matrigel plug assay (Popa, Mirel-Adrian et al. "Dihydrotestosterone induces pro-angiogenic factors and assists homing of MSC into the cardiac tissue." Journal of Molecular Endocrinology vol. 60,1 (2018): 1-15. doi:10.1530/JME-17-0185). As suggested, we plan to use a similar animal model of cardiovascular disease and evaluate the therapeutic efficacy of DHT-treated EPCs. This will be the topic of another paper.

  • It's recommended to add a cell scratch assay to demonstrate the effect of DHT on EPCs.

Answer: Thank you for your important comment. We have realized that we did not explain well enough that the use of our patented device is equivalent to a "scratch assay". We have revised thoroughly the description of the device, and of the procedure explaining that the method stands for an improved “scratch assay” ( Materials and Methods of the revised manuscript (page 15, lines 540-551).

  • In Figure 4, please clearly indicate which bands belong to the EPC group and which to the EPC+DHT group. Additionally, the molecular weight information is unclear; please replace it with an electronic version instead of a handwritten format.

Answer: Good suggestions. In the revised manuscript, we have indicated the bands that belong to EPC and EPC+DHT groups. Handwritten molecular weight has been replaced by an electronic version.

  • In the statistical results of the Western blot experiment, why was the EPC control group not normalized to 1?

Answer: In our initial analysis, we presented the data as mean normalized samples. This approach was chosen to allow for a straightforward comparison of raw densitometry values across all samples. However, we understand your point about normalizing the EPC control group to 1. Consequently, we have revised our analysis and in the revised figure the EPCs control group was normalized to 1. Thanks for your insightful comment.

We thank the reviewer for the constructive criticism and the attention paid to our paper which helped us to improve the revised manuscript. We hope that the revised form will meet the criteria of IJMS.

The authors

Reviewer 3 Report

Comments and Suggestions for Authors

The aim of this manuscript is to investigate the role of DHT in human endothelial progenitor cell proliferation and migration. The authors utilized different concentrations of DHT on human EPCs and demonstrated that DHT could stimulate the proliferation and migration of human EPCs in vitro. They also characterized several protein and gene markers for potential underlying mechanisms.

While this study is well-designed and offers detailed mechanistic insights through various methods, there are several concerns that need to be addressed before considering publication:

  1. The effect of androgens on EPCs has been studied for many years, and the findings in this manuscript may not be considered particularly novel compared to others. It is essential to clarify the unique contribution of this study compared to prior research.
  2. The purpose of presenting Figure 1 and the first paragraph of the Results section is not very clear and may confuse readers.
  3. The figure legends for each figure need extensive revision, especially with clearer abbreviations. This will facilitate better understanding for readers and ensure the figures are effectively interpreted.
  4. The concentration of DHT in each figure should be demonstrated more clearly. It appears that the investigators used 30nM in several studies but it is not clear in others. Ensuring consistency and clarity in reporting DHT concentrations across all experiments is essential.
  5. Based on the concentration curve in Figure 2, it seems that 10nM DHT showed the most significant effect on EPCs. What is the reason for using 30nM in some of the subsequent studies? Providing justification for the choice of DHT concentration in different experimental conditions would strengthen the manuscript's scientific rigor.

Author Response

We thank the reviewer for excellent comments and constructive criticism of our paper which helped us to improve the revised manuscript.

Please find below a point-by-point answer to the comments:

Comment 1: The effect of androgens on EPCs has been studied for many years, and the findings in this manuscript may not be considered particularly novel compared to others. It is essential to clarify the unique contribution of this study compared to prior research.

Answer: The comment was well taken, and we have revised the manuscript to emphasize the unique contribution of our study in the context of existing research on the effect of androgens on EPCs. This revision can be found at page 11, lines 333-340.

Comment 2: The purpose of presenting Figure 1 and the first paragraph of the Results section is not very clear and may confuse readers.

Answer: As suggested, we have revised the first paragraph of the Results section and the legend to Figure 1 to mention that for our experiments we have used human umbilical blood-derived EPCs that being a novel experimental setup needed to be well characterized.

Comment 3: The figure legends for each figure need extensive revision, especially with clearer abbreviations. This will facilitate better understanding for readers and ensure the figures are effectively interpreted.

Answer: The reviewer is right. In the revised manuscript we have corrected the legends for each figure to ensure that abbreviations are clearly explained.

Comment 4: The concentration of DHT in each figure should be demonstrated more clearly. It appears that the investigators used 30nM in several studies, but it is not clear in others. Ensuring consistency and clarity in reporting DHT concentrations across all experiments is essential.

Answer: We have ensured that in the revised manuscript, the concentration of DHT is clearly shown in each legend. We have consistently used 30nM in our studies and have added this information to the legends.

Comment 5: Based on the concentration curve in Figure 2, it seems that 10nM DHT showed the most significant effect on EPCs. What is the reason for using 30nM in some of the subsequent studies? Providing justification for the choice of DHT concentration in different experimental conditions would strengthen the manuscript's scientific rigor.

Answer:  We are sorry that the sentence was confusing. In the revised manuscript we made it clear that we have assessed the effect of various concentrations of DHT (1, 30, 100 and 1000 nM) on EPCs. We didn’t use 10 nM DHT. We have provided justification for using 30 nM DHT in our further experiments, namely ‘to be in line to the physiological conditions….’ (page 3 lines 96-98).

We thank to the reviewer for the time and attention paid to our manuscript and we hope that the present form will meet the requirements of IJMS.

The authors

Reviewer 4 Report

Comments and Suggestions for Authors

The manuscript titled, Dihydrotestosterone augments the angiogenic and migratory potential of human endothelial progenitor cells by an androgen-dependent mechanism’ is interesting and well-written. The authors mainly revealed that DHT potentiates androgen receptor-mediated angiogenesis by increasing the proliferation, migration, and upregulation of proangiogenic factor secretion by the EPCs. They also showed that DHT facilitates the recruitment of EPCs in the cardiac tissue that can be used to regenerate damaged endothelial cells in the heart.  Although the whole manuscript looks promising and contains a lot of data, I have several concerns as follows.

1.     The authors need to mention the concentration/dose of flutamide used in this study's different experiments.

 2.     One of the major limitations of this study is the lack of a particular disease model. According to this study, DHT works as a proangiogenic factor. Therefore, based on the disease scenario, DHT can be protective (e.g., wound healing) or deleterious (e.g., cancer, diabetic retinopathy, atherosclerosis). Considering this scenario the authors need to explain in detail to support their findings in the discussion section.

 3.     Why did the author use cardiac tissue to observe the recruitment of EPC? Other tissues, including the kidney and skeletal muscle, express the androgen receptors, so the authors need to justify their selection of cardiac tissue over other tissues.

Author Response

We thank the reviewer for excellent comments and constructive criticism of our manuscript which we hope improved our revised manuscript.

Please find below a point-by-point answer to the comments:

Comment 1: The authors need to mention the concentration/dose of flutamide used in this study's different experiments.

Answer: In the revised manuscript we have included the specific concentration/dose of Flutamide used in each experiment.

Comment 2: One of the major limitations of this study is the lack of a particular disease model. According to this study, DHT works as a proangiogenic factor. Therefore, based on the disease scenario, DHT can be protective (e.g., wound healing) or deleterious (e.g., cancer, diabetic retinopathy, atherosclerosis). Considering this scenario the authors need to explain in detail to support their findings in the discussion section.

Answer: We appreciate the reviewer’s excellent comment. First, we added at the “Limitations of the study” the lack of disease model. Second, in the revised manuscript we emphasize that care should exercise and the use of DHT should be done according to the disease scenario. We have clarified this point in our revised manuscript at page 15, lines 477-479.

Comment 3.  Why did the author use cardiac tissue to observe the recruitment of EPC? Other tissues, including the kidney and skeletal muscle, express the androgen receptors, so the authors need to justify their selection of cardiac tissue over other tissues.

Answer to Comment: The second sentence in the Discussion section was revised to explain that in the heart tissue repair EPCs have the main role to replace damaged endothelial cells. For this reason, we chose to use cardiac tissue to study the recruitment of EPCs.

We thank the reviewer for the time and attention paid to our manuscript. The excellent comments helped us to improve our paper and we hope that the revised manuscript will meet the criteria of IJMS. 

The authors

Round 2

Reviewer 2 Report

Comments and Suggestions for Authors

I have no further concerns.

Reviewer 3 Report

Comments and Suggestions for Authors

The authors response all the reviewer's comments with modification of the manuscript. The manuscript should be consider to be published.